# Dynamic Parameter Memory: Temporary LoRA-Enhanced LLM for Long-Sequence Emotion Recognition in Conversation

## Abstract

Recent research has focused on applying speech large language model (SLLM) to improve speech emotion recognition (SER). However, the inherently high frame rate in speech modality severely limits the signal processing and understanding capabilities of SLLM. For example, a SLLM with a 4K context window can only process 80 seconds of audio at 50Hz feature sampling rate before reaching its capacity limit. Input token compression methods used in SLLM overlook the continuity and inertia of emotions across multiple conversation turns. This paper proposes a Dynamic Parameter Memory (DPM) mechanism with contextual semantics and sentence-level emotion encoding, enabling processing of unlimited-length audio with limited context windows in SLLM. Specifically, DPM progressively encodes sentence-level information and emotions into a temporary LoRA module during inference to effectively "memorize" the contextual information. We trained an emotion SLLM as a backbone and incorporated our DPM into inference for emotion recognition in conversation (ERC). Experimental results on the IEMOCAP and MELD datasets show that DPM significantly improves the emotion recognition capabilities of SLLM when processing long audio sequences, achieving state-of-the-art performance.

## 1 Introduction

Emotions play a crucial role in human communication. Speech Emotion Recognition (SER) is an important tool for providing user emotional information to intelligent systems (Gross & Muñoz, 1995; Gross et al., 2019). It has wide applications in smart robots, automated call centers, and remote education (Li et al., 2019; Akçay & Oğuz, 2020). Emotion recognition in conversation (ERC) extends SER to enhance emotion recognition capabilities based on conversation history.

Traditional emotion recognition relies on architectures with pretrained representations and well-designed downstream classification networks (Mai et al., 2024; Chen et al., 2022c). Recently, speech large language model (SLLM), which incorporates speech understanding into language models, has significantly pushed the boundaries of various speech-related tasks. In particular, their potential for SER has garnered increasing attention (Yang et al., 2024a; Bukhari et al., 2024).

However, the high frame rate characteristic of the audio modality poses significant challenges for SLLM in processing emotional dialogues, as the limited context windows of language models often fail to accommodate sufficiently long audio sequences. For example, a SLLM with a 4K token context window processing audio representations at 50Hz sampling rate can only handle about 80 seconds of content, far insufficient for real-world conversations or meeting recordings, as shown in Figure 1. Input token compression methods (Xu et al., 2025; Yang et al., 2024b) have been proposed to address the above limitation. However, these approaches often overlook emotional continuity and transfer across multi-turn dialogues.

Additionally, even training models with longer context windows to alleviate capacity limitations (Ding et al., 2025; Yang et al., 2024a), conventional attention mechanisms (Vaswani et al., 2017) suffer from quadratic computational complexity with respect to input sequence length, rendering them inefficient for long audio inputs.

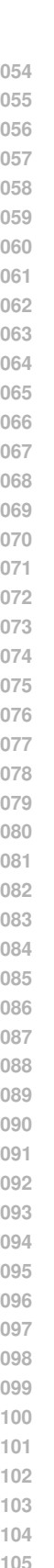
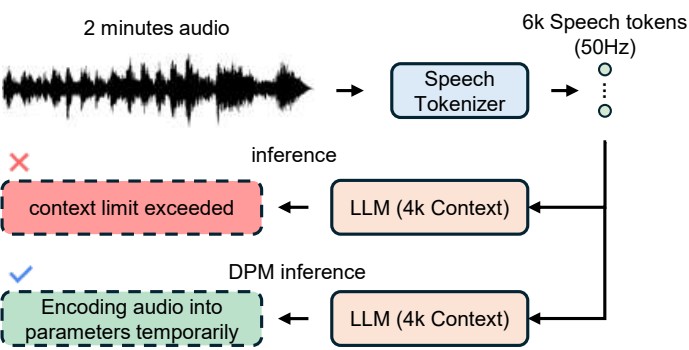

Figure 1: When using a SLLM with a 4k context window to infer a 2-minute audio sequence (at 50Hz), the input exceeds the limit. In contrast, DPM enables successful inference by encoding the audio into parameters sentence by sentence.

Whether through adapting inputs to fit existing models or modifying models to accommodate longer inputs, these approaches ultimately encounter fundamental limitations when dealing with sufficiently long audio sequences. Their approach eases symptoms, but does not tackle the core problem.

We introduce the Dynamic Parameter Memory (DPM), a novel inference mechanism designed to process unlimited-length emotional audio dialogues within a limited context window. While the concept of updating parameters at inference time, as explored by Temporary LoRA (Wang et al., 2024) for text, provides a promising direction, its direct application to speech is non-trivial due to the high temporal density and unique structure of emotional expression in audio. To bridge this gap, our core contribution is a two-fold approach: first, we develop an emotion-aware SLLM specifically pre-trained to comprehend emotional nuances in audio sequences. Second, we build the DPM mechanism upon this foundation, which iteratively distills sentence-level audio information and emotional context into a temporary LoRA module. This allows our method to maintain a dynamically updated "memory" of the conversation, achieving a deep, context-aware understanding of long dialogues.

To verify DPM's effectiveness, we conduct extensive ERC experiments on two standard benchmarks: IEMOCAP (Busso et al., 2008) and MELD (Poria et al., 2018). Experimental results consistently demonstrate that, within the same emotion-aware SLLM framework, the simple integration of the DPM inference mechanism substantially improves emotion recognition performance, particularly on long audio sequences. Furthermore, our method shows robust gains across samples of various lengths, confirming its general applicability.

Furthermore, our proposed emotion SLLM itself outperforms traditional classifier approaches, demonstrating the unique advantages of autoregressive structures in audio sequence understanding. Comparative experiments show that our DPM-enhanced method achieves state-of-the-art (SOTA) performance on both datasets.

The main contributions are summarized as follows:

- We design and train an effective emotion SLLM that comprehensively outperforms traditional classifier approaches, establishing a solid foundation for the application of the DPM mechanism.

- We propose the Dynamic Parameter Memory (DPM) mechanism, a novel inference strategy that fundamentally addresses the challenge of processing long emotional speech in SLLM, enabling the handling of unlimited-length audio with linear computational complexity under a limited context window.

- We conduct extensive experiments on the IEMOCAP and MELD datasets, demonstrating that our DPM-enhanced SLLM achieves new state-of-the-art results on both benchmarks.

- We provide an in-depth analysis of DPM's stepping strategies, revealing a critical trade-off between performance and efficiency and reinforcing the value of using semantic units for memory updates.

## 2 RELATED WORK

### 2.1 DOWNSTREAM NETWORKS FOR SER

Leveraging representations from pretrained models followed by carefully designed downstream networks and classifiers has been the mainstream approach in SER. Pretrained models such as wav2vec (Baevski et al., 2020), HuBERT (Hsu et al., 2021), WavLM (Chen et al., 2022b), and Whisper (Radford et al., 2022) have utilized large amounts of unsupervised and supervised speech data to transform audio into abstract, high-dimensional representations.

Despite their differences in discrete or continuous representations, these models have gained generalization capabilities through task-agnostic pretraining strategies, enabling their applicability across a wide range of downstream tasks. Therefore, speech emotion recognition tasks can be developed based on pretrained representations.

However, SER tasks typically avoid directly fine-tuning these pretrained models for two reasons. First, task-specific fine-tuning may compromise the generalization capabilities gained from pretraining on vast amounts of audio data. Second, these pretrained models are often computationally expensive to fine-tune due to their large size. A more appropriate approach involves attaching trainable downstream networks to frozen pretrained models. By carefully designing downstream networks that address the specific characteristics of speech emotion understanding and fine-tuning them on emotion-labeled speech data, researchers can effectively transfer the general speech knowledge from pretrained models to SER, ultimately making emotional judgments through classifiers.

### 2.2 SLLM FOR SER

SLLM has emerged as powerful tools for speech understanding tasks, including emotion recognition. Unlike traditional approaches that design different downstream networks and classifiers for various audio understanding tasks, SLLM integrates diverse downstream tasks into a unified framework, benefiting from the transfer of intrinsic reasoning capabilities and robust semantic understanding acquired by large language models through training on massive text corpora.

SLLM still requires speech pretrained models as their entry point for audio comprehension. This necessity arises from two key factors: first, the high frame rate of raw waveforms makes them unsuitable as direct inputs to LLMs; second, pretrained model representations possess abstract semantic capabilities that facilitate alignment between audio and semantic knowledge within LLMs. An Adapter typically connects the pretrained model and LLM, serving the dual purpose of dimension alignment and semantic alignment. The continuous audio representations output by the Adapter are temporally concatenated with text instruction representations obtained through the tokenizer, forming the prefix for LLM autoregressive inference.

Recent advancements in SLLM have demonstrated promising results for SER. Models such as Qwen2-Audio (Yang et al., 2024a) and SELM (Bukhari et al., 2024) showcase the effectiveness of extending language models to understand emotional cues in speech. These models utilize continuous audio representations as input, enabling LLMs to develop audio emotion understanding capabilities through training on substantial emotion-labeled speech data, while also being able to describe audio emotions using natural language.

However, SLLM faces significant challenges when processing long audio sequences, particularly in ERC. Due to the limited context windows of LLM, the high frame rate of audio data means that even a few minutes of speech can exceed the token capacity of most LLM, necessitating innovative solutions to address this fundamental limitation.

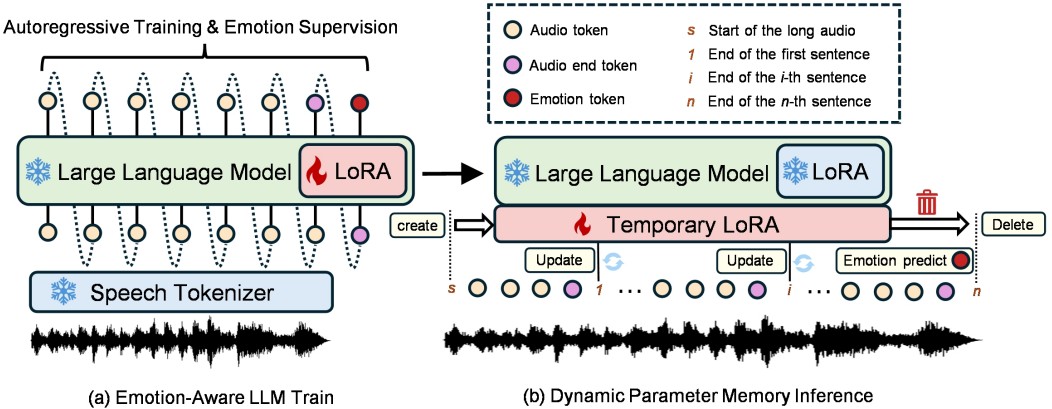

Figure 2: On the left is the training of the emotion SLLM foundation, which includes autoregressive audio training and emotion supervision. On the right, it shows how DPM enables infinite-length audio inference based on the emotion SLLM.

## 2.3 ENABLING LLM TO PROCESS LONGER SEQUENCES

Processing long sequences remains a persistent challenge in LLM research. SLLM faces even greater difficulties when handling extended audio content due to the inherently high frame rate of audio features.

Two main approaches have emerged to address this limitation. The first involves training LLM with expanded context windows to accommodate longer inputs, as seen in models like Kimi (Ding et al., 2025) and Qwen2.5 (Yang et al., 2024a). The second approach focuses on compressing inputs to fit within limited context windows, either by utilizing extremely low frame rate codecs as LLM inputs (for example, Mimi codec (Défossez et al., 2024) can compress 1 second of 24kHz audio to just 12.5 frames), or by implementing multi-scale transformer architectures where one decoder summarizes and compresses the input while another decoder performs autoregressive prediction based on the compressed representation (Xu et al., 2025).

Despite these advancements, most existing methods still encounter fundamental limitations when processing extremely long audio sequences. As input lengths increase beyond a certain threshold, all existing approaches ultimately encounter performance ceilings.

Murph et al. proposed a method (Murph et al., 2024) that condenses historical information into fixed-length embeddings, which fundamentally alleviates the difficulty of LLMs processing long audio, but potentially suffers from information loss. Wang et al. introduced Temporary LoRA (Wang et al., 2024), which progressively encodes historical text information into parameter space, effectively addressing the challenges of limited-window LLM in understanding long texts. However, this approach lacks validation in other modalities, particularly in the audio domain where high frame rates always present significant challenges.

## 3 METHODOLOGY

The training process of our emotion SLLM and the inference process of DPM are shown in Figure 2. We first train an emotion SLLM as a foundation model with audio understanding and emotion reasoning capabilities for DPM inference. The implementation details are described below, using our setup for the IEMOCAP dataset (four-class classification) as the primary example.

### 3.1 EMOTION SLLM

As shown in the left side of Figure 2, the emotion SLLM takes the output of a speech tokenizer as input and autoregressively predicts the next audio token (yellow tokens). It outputs an end identi-

fier (purple token) at the end of a sentence, followed by an emotion identifier (red token) for that sentence.

To enable the LLM to process these discrete audio codes, we first expand the vocabulary size of the LLM tokenizer to accommodate the speech tokenizer's codebook size, represented as $\langle \text{audio}\_i \rangle$. We also add a token representing the end of the current sentence, denoted as $\langle \text{audio\_end} \rangle$, and four emotion identifiers represented as $\langle \text{emo\_hap} \rangle$, $\langle \text{emo\_sad} \rangle$, $\langle \text{emo\_ang} \rangle$, and $\langle \text{emo\_neu} \rangle$.

The model input-output structure adjustment to the tokenizer embedding layer can be formulated as:

$$\mathbb{R}^{|V_{\text{text}}|} \to \mathbb{R}^d \tag{1}$$

$$\mathbb{R}^{|V_{\text{text}}|+|\langle \text{audio}\_i \rangle|+1+4} \to \mathbb{R}^d \tag{2}$$

where $|V_{\text{text}}|$ is the original text vocabulary size, $|\langle \text{audio}\_i \rangle|$ is the speech tokenizer codebook size, 1 represents one sentence end identifier, 4 represents the number of emotion identifiers, and $d$ is the LLM embedding dimension. Equation 1 is before the adjustment, and Equation 2 is after the adjustment.

We use LoRA (Hu et al., 2021) to train the SLLM. Before training, we preprocess the training data. For a long audio sample, we first convert the audio into discrete codes using a speech tokenizer. Then, we insert $\langle \text{audio\_end} \rangle$ at the end of each sentence, followed by the corresponding emotion identifier. For example, for a 5-minute audio consisting of 50 sentences, we would insert 50 $\langle \text{audio\_end} \rangle$ tokens and 50 corresponding emotion identifiers at the end of each sentence.

The training includes two core objectives: 1) audio understanding ability and 2) emotion classification ability.

### 3.1.1 AUDIO AUTOREGRESSIVE TRAINING

We train the model's next token prediction to ensure its understanding of audio. For each sentence, we take $n_o$ tokens before $\langle \text{audio\_end} \rangle$ (including $\langle \text{audio\_end} \rangle$) as the training target for autoregressive prediction. We take $n_p$ tokens before these as the prefix for predicting the next $n_o$ tokens, using teacher forcing for autoregressive training. The loss for these $n_o$ tokens is denoted as $\mathcal{L}_a$.

$$\mathcal{L}_a = -\frac{1}{n_o} \sum_{k=1}^{n_o} \log P \left( x_{T-n_o+k} \,\big|\, x_{T-n_o-n_p:T-n_o+k-1} \right) \tag{3}$$

where $T$ represents the position of $\langle \text{audio\_end} \rangle$, $x$ is a sequence of audio token after preprocessing.

Set the total sequence length of the current sentence and its history is $n$, for boundary cases:

$$\begin{cases} n_p = 0, \quad n_o = n & \text{if } n < n_o \\ n_p = n - n_o, \quad n_o = n_o & \text{if } n_o \le n < n_o + n_p \end{cases} \tag{4}$$

Through audio autoregressive training, the emotion SLLM learns to autoregressively predict the next audio token when receiving an audio token and predict $\langle \text{audio\_end} \rangle$ at appropriate times, gaining audio understanding ability.

### 3.1.2 EMOTION SUPERVISION TRAINING

We also train the model's emotion classification ability. For each sentence ending, we take the preceding $n_q$ tokens (including $\langle \text{audio\_end} \rangle$) as the prefix for predicting the emotion identifier. Notably, for the emotion SLLM's output, we constrain the logits from the original size of $|V\text{text}| + |\langle \text{audio}\_i \rangle| + 1 + 4$ to just 4 (position of emotion identifiers) to increase the stability of emotion prediction, constrained logits denoted as $\hat{p}$. We then calculate the cross-entropy loss $\mathcal{L}_e$ using the constrained logits and the true emotion identifier.

$$\mathcal{L}_e = -\sum_{k=1}^{4} y_k \log \hat{p}_k \tag{5}$$

where $y_k$ is the real emotional identifier one-hot vector.

For boundary cases:

$$n_q = n \quad \text{if} \quad n < n_q \tag{6}$$

The overall training loss is:

$$\mathcal{L} = \frac{1}{2}(\mathcal{L}_a + \mathcal{L}_e) \tag{7}$$

## 3.2 DPM INFERENCE

As shown on the right side of Figure 2, we freeze the emotion SLLM and its LoRA module. For each long audio sequence sample, at the beginning of its emotion inference (marked as time $s$ in the figure), we create a temporary LoRA module specifically for this audio sequence.

Unlike previous approaches that input the entire sequence into the SLLM at once, we process information sentence by sentence, which is the key to our ability to process infinitely long sequences. At the end of the $i$-th sentence (time $i$ in the figure), we take its preceding $n_r$ tokens as the prefix for autoregressive prediction, and predict the audio tokens of the $(i + 1)$-th sentence.

$$\hat{x}_{i+1} = \text{SLLM}\left(x_{T-n_r:T}\right) \tag{8}$$

where $x$ represents inference audio tokens output by speech tokenizer. $\hat{x}_{i+1}$ represents the predicted tokens for the $(i + 1)$-th sentence, $x_{T-n_r:T}$ denotes the $n_r$ speech tokens preceding the end position $T$ of the $i$-th sentence.

At this stage, the emotion SLLM utilizes its general audio understanding capabilities acquired during pretraining to predict the tokens of the $(i + 1)$-th sentence. However, for long audio sequences, there might be discrepancies between the general knowledge and the specific context of the current conversation.

For instance, during pretraining, the emotion SLLM may have developed a relatively neutral understanding of audio. During DPM inference, if the first sentence transcription is "I've been waiting for you for an hour," the emotion SLLM might predict that the second sentence would be "I feel tired" based on its general knowledge of audio understanding and emotional patterns. However, if this conversation actually involves an argument, the ground truth for the second sentence might be "I feel angry." This presents a critical challenge: how to "remember" the content of the second sentence, and how to propagate the emotion of anger from this long sample to the inference of subsequent sentences to aid in the final emotion judgment.

Naturally, we address this by calculating the loss between the ground truth of the second sentence and the model's prediction of the second sentence, which serves our purpose effectively. When we update the temporary LoRA parameters using this loss, we essentially encode both the content and the emotional context of the second sentence into the parameter space.

We calculate the autoregressive loss $\mathcal{L}_t$ between the predicted and true audio tokens of the $(i+1)$-th sentence, and update the temporary LoRA parameters.

$$\mathcal{L}_t = -\frac{1}{m} \sum_{k=1}^{m} \log P\left(x_{T-m+k} \mid x_{T-m-n_r} : x_{T-m+k-1}\right) \tag{9}$$

where $m$ is the number of tokens in the predicted $(i + 1)$-th sentence.

Through this method, we encode the information of the $(i + 1)$-th sentence into the parameter space of the temporary LoRA. For a 5 minute audio with 50 sentences, the temporary LoRA module will be updated 49 times, encoding the information of the long audio sequence into the parameter space sentence by sentence. If we denote the maximum sentence token count in the current long audio sample as $n_{\max}$ and the SLLM's window limit as $n_{\text{limit}}$, then DPM can process unlimited-length audio sequences as long as:

$$n_{\text{limit}} \geq n_{\max} + n_r \tag{10}$$

---

**Algorithm 1** DPM inference

---
1: **Freeze** SLLM and its LoRA module
2: **for** sample in test_set **do**
3:    temp_lora ← SLLM.create_templora()
4:    **for** sentence in sample.sentences **do**
5:      **if** $T$ is undefined **then**
6:        $T$ ← sentence.end_time
7:        **continue**
8:      **end if**
9:      prefix ← sample.tokens$[T - n_r : T]$
10:     pred_tokens ← SLLM.infer(prefix)
11:     loss ← loss_fn(pred_tokens, sentence.tokens)
12:     temp_lora.zero_grad()
13:     loss.backward()
14:     temp_lora.step()
15:     T ← sentence.end_time
16:    **end for**
17:    delete(temp_lora)
18: **end for**

---

For boundary cases:

$$n_r = n \quad \text{if} \quad n < n_r \tag{11}$$

After the last sentence of the long audio sequence, we update the temporary LoRA parameters one last time and predict an emotion token as the final emotion classification for the entire sample. We then discard the temporary LoRA module since it stores the information specific to the current sample and should not interfere with future sample inference. The pseudo code of DPM inference is shown as Algorithm 1.

## 4 EXPERIMENTS

### 4.1 DATASETS

**IEMOCAP** The IEMOCAP dataset (Busso et al., 2008) consists of 151 video recordings split into 5 sessions. Keeping in line with previous works (Li et al., 2025; Kieu et al., 2025), we perform a four-class classification task where "angry", "happy", "sad", and "neutral" categories are considered (with the excited category merged into happy). As a result, a total of 5531 utterances spanning four emotion categories are collected. For data partitioning, we adopt the popular "LOSO" (Leave-One-Session-Out) strategy.

**MELD** The MELD (Poria et al., 2018) is a multi-speaker conversational dataset derived from the TV series *Friends*. It comprises 1,433 dialogues from 407 different speakers. The dataset is officially partitioned into training (1,039 dialogues), validation (114 dialogues), and test (280 dialogues) sets. Each utterance is annotated with one of seven emotion categories: "angry", "disgust", "sad", "joy", "neutral", "surprise", and "fear". We adhere to the official data splits for our experiments.

### 4.2 EXPERIMENT SETUP

We use Llama2-7B (Touvron et al., 2023) with a 32k context window as our base LLM. We choose the 32k context window because we focus on comparing DPM inference with standard inference, and Llama2's default 4k context cannot handle long audio, making comparison difficult. For speech tokenization, we use CosyVoice2 (Du et al., 2024) tokenizer with a 25Hz single codebook. Both the emotion SLLM's LoRA and the temporary LoRA used in DPM adopt a LoRA rank of 64 and a LoRA alpha of 64, activating 0.16B parameters. Both the emotion SLLM's LoRA and the temporary LoRA in DPM are applied to all linear layers of the LLM.

To establish a strong baseline and isolate the gains from our SLLM's autoregressive structure, we compared our model against a traditional classifier-based approach. This baseline model processes

Table 1: Ablation study on IEMOCAP and MELD. We compare our full model (**SLLM-DPM**) against its variants under two settings: (1) using only complete dialogue samples and (2) using samples of all lengths.

| Method | IEMOCAP | | | MELD |
|---|---|---|---|---|
| | WA (%) | UA (%) | WF1 (%) | WF1 (%) |
| *Setting 1: On complete dialogue samples* | | | | |
| **SLLM-DPM** | **79.38** | **79.62** | **79.34** | **51.22** |
| SLLM | 72.82 | 73.34 | 73.58 | 47.90 |
| Classifier | 70.96 | 70.51 | 70.64 | 44.78 |
| *Setting 2: On samples of all lengths* | | | | |
| **SLLM-DPM** | **75.38** | **74.67** | **74.99** | **52.82** |
| SLLM | 73.15 | 71.98 | 73.12 | 52.53 |
| Classifier | 66.19 | 63.40 | 65.22 | 46.84 |

the speech tokenizer's output codes through a trainable Transformer encoder followed by a classification head. For training, the learning rate for the emotion SLLM was set to 5e-5, while the classifier baseline used 1e-4. For our DPM inference, the temporary LoRA module was updated with a learning rate of 5e-5. All models were trained for 20 epochs, and all experiments were conducted on a single NVIDIA A800 GPU.

The prefix length for DPM inference (the number of tokens preceding the final emotion token used for prediction) was tuned as a hyperparameter for each dataset. We set it to 1024 for IEMOCAP and 256 for MELD. This choice aligns with the datasets' characteristics: IEMOCAP features significantly longer dialogues (avg. 64.96 utterances) compared to MELD (avg. 9.80 utterances). For the lengthy and complex conversations in IEMOCAP, a larger prefix provides a more robust "recent context" to anchor the extensive parametric memory accumulated by the temporary LoRA. Conversely, the more concise dialogues in MELD require a shorter prefix for an effective and efficient final prediction.

### 4.3 EXPERIMENTAL RESULTS AND ANALYSIS

#### 4.3.1 ABLATION STUDY

As shown in Table 1, we present the results of the ablation study. The upper part shows the emotion recognition performance on complete dialogue samples, while the lower part shows the performance on emotion recognition without length constraints, covering lengths from single sentences to full dialogues, aiming to provide a more comprehensive evaluation of DPM's performance when handling different lengths.

*SLLM-DPM* represents DPM inference on emotion SLLM, *SLLM* represents performing a one-time autoregressive inference on emotion SLLM without using DPM, and *Classifier* represents using a classifier after the speech tokenizer instead of LLM for training and inference. For complete dialogue emotion experiments, we show emotion recognition results only from complete dialogue audio. We observe that for understanding emotions in long audio sequences, simply changing the inference method to DPM significantly outperforms direct inference with emotion SLLM (inputting the complete audio sample at once and then inferring the emotion identifier), with a 6.56% improvement in WA on IEMOCAP. A similar trend is observed on the MELD dataset, where DPM achieves a 3.32% gain in WF1. Compared to the classifier baseline, our full model shows an 8.42% improvement in WA on IEMOCAP and a 6.44% improvement in WF1 on MELD.

This clearly demonstrates the advantage of DPM inference for emotion understanding in long audio sequences. In contrast to processing the entire sequence at once (similar to human "skim-reading"), which can lead to missing critical emotional information, DPM incrementally processes and retains information block by block, resembling a "line-by-line" reading strategy. This enables more precise capture of emotional information in long audio sequences.

Table 2: Comparison with recent models for ERC on IEMOCAP and MELD datasets. The best results are in bold.

| Method | IEMOCAP | | | MELD |
|---|---|---|---|---|
| | WA (%) | UA (%) | WF1 (%) | WF1 (%) |
| ESA-CRF (Chen et al., 2022a) | 73.17 | 74.47 | - | - |
| MER-HAN (Zhang et al., 2023) | 73.33 | 74.20 | 73.66 | 40.50 |
| Mi-CGA (Kieu et al., 2025) | - | - | 73.77 | - |
| MER-TL (Padi et al., 2022) | 75.76 | 76.07 | - | - |
| GatedxLSTM (Li et al., 2025) | 76.34 | - | 75.97 | - |
| MERITS-L (Dutta & Ganapathy, 2025) | - | - | 77.95 | 49.32 |
| **DPM (ours)** | **79.38** | **79.62** | **79.34** | **51.22** |

Even when extending the duration range of the test audio to cover all length ranges, DPM inference still maintains a significant advantage, with WA exceeding direct emotion SLLM inference by 2.23% on IEMOCAP, and showing a gain on MELD as well. This demonstrates the general applicability of DPM inference.

The emotion SLLM functions like a long-term memory system in the brain, responsible for understanding general emotional knowledge and basic audio information. The temporary LoRA simulates short-term working memory, specifically storing contextual information of the current audio. This division of labor is well-suited for audio modality (with large temporal spans), thus maintaining performance improvements.

When using emotion SLLM for direct inference without DPM, our method outperformed traditional classifier approaches in both settings and across both datasets, indicating the unique advantage of the autoregressive structure over classifiers in understanding emotions in audio sequences.

It's worth noting that since DPM inference computation is proportional to the number of sentences, DPM inference achieves linear computational complexity. When audio sequence length increases, inference computational complexity grows linearly. In addition to these main results, we conducted further experiments to analyze the DPM stepping strategies and the impact of context window size on performance. These detailed analyses are presented in the Appendix.

### 4.3.2 COMPARISON STUDY

We compared our model with recent models for ERC in Table 2. To ensure fairness, all compared models used IEMOCAP's conventional four-class method. For multimodal methods, we selected results from their audio modality. MERITS-L (Dutta & Ganapathy, 2025) leverages LLMs to assist with the ERC task, while other methods improve on traditional classifier approaches. The performance of SLLM-based emotion recognition generally surpassed traditional classifier approaches, confirming our conclusions from the ablation study. With the same experimental settings, our method achieved SOTA results on both the IEMOCAP and MELD datasets. This demonstrates the effectiveness of our designed emotion SLLM combined with DPM inference for capturing emotions in long audio sequences.

## 5 CONCLUSIONS

The inherently high frame rate in the speech modality severely limits the emotion understanding capabilities of SLLM. This paper proposes a Dynamic Parameter Memory (DPM) mechanism with contextual semantics and sentence-level emotion encoding, enabling processing of unlimited-length audio with limited context windows in SLLM. Our method progressively encodes sentence-level information and emotions into a temporary LoRA module during inference, effectively "memorizing" contextual information. We trained an emotion SLLM as the basis for DPM inference. We validated our method on ERC. Experimental results on the IEMOCAP dataset show that DPM significantly improves the emotion recognition capabilities of SLLM when processing long audio sequences, achieving state-of-the-art performance.

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

## A  APPENDIX

### A.1  ANALYSIS OF DPM STEPPING STRATEGIES

Our primary implementation of DPM employs a sentence-by-sentence progression, which, while intuitive, may not be the most efficient. To investigate this further, we conducted an additional experiment on the MELD dataset to explore the trade-off between performance and efficiency under

different stepping strategies. We compare our original sentence-based approach against a fixed-stride stepping strategy during DPM inference. The training process remains identical, only the mechanism for updating the temporary LoRA module during inference is changed from advancing by one semantic sentence to advancing by a fixed number of tokens. We tested stride lengths of 64, 128, 256, 512, and 1024 tokens.

Table 3: Impact of DPM stepping strategies on performance on the MELD dataset.

| Stepping Strategy | WF1 (%) |
|---|---|
| Sentence-by-sentence (Baseline) | **51.22** |
| Fixed-Stride (64 tokens) | 49.61 |
| Fixed-Stride (128 tokens) | 49.64 |
| Fixed-Stride (256 tokens) | 47.63 |
| Fixed-Stride (512 tokens) | 43.25 |
| Fixed-Stride (1024 tokens) | 37.87 |

The results, presented in Table 3, reveal several key insights. Firstly, the sentence-by-sentence strategy outperforms all fixed-stride settings. This suggests that semantic boundaries, such as sentences, provide more meaningful and coherent chunks of information for the DPM to memorize than arbitrary fixed-length segments. This aligns with the intuition that emotions are often expressed and evolve at the sentence level.

Secondly, a clear trend of performance degradation is observed as the fixed stride becomes more aggressive. The best performance among fixed-stride settings is achieved with a stride of 128 tokens (49.64% WF1), after which performance drops significantly. This indicates that there is an optimal "glimpse" length for the DPM mechanism. Interestingly, the average utterance duration in the MELD dataset is 3.22 seconds, which corresponds to approximately 80 tokens at a 25Hz feature frame rate. The fact that the best-performing fixed stride (128 tokens) is in a similar range further reinforces the hypothesis that a sentence-like unit is the most effective quantum for emotional context updates.

Finally, we analyze the efficiency trade-off. While the sentence-by-sentence approach is most accurate, fixed-stride stepping offers substantial efficiency gains. For instance, a stride of 1024 tokens improves inference speed by a factor of 12.72 compared to the sentence-by-sentence approach, but this comes at a significant cost, with the WF1 score dropping by 13.35% (from 51.22% to 37.87%). A more moderate stride of 512 tokens improves inference speed by a factor of 6.26, while performance decreases by 7.97%. This analysis highlights a clear trade-off, allowing practitioners to choose a strategy that balances the need for accuracy with computational constraints.

## A.2 DOES TRUNCATED CONTEXT AFFECT LLM'S EMOTION UNDERSTANDING?

Ablation and comparative experiments were conducted on LLM with 32k context windows to verify the advantages of DPM over traditional one-time autoregressive inference and traditional classifiers. Using 32k context LLM ensured fairness in input information, as the official Llama2 version has only a 4k context window, insufficient to accommodate complete long audio information at once.

The ablation experiments implicitly established an important standpoint: when using LLM for emotion understanding of long audio, complete context information is beneficial for understanding.

In our experiment, we used both 4k and 32k context versions of Llama as base models for our speech emotion LLM training, following the same procedures described earlier. We then input complete audio dialogue sequences into the SLLM for one-time emotion inference without DPM. Obviously, the 4k context experimental setting cannot fully accommodate long audio inputs, we truncated the audio sequence from the end to 4k to enable one-time inference.

As shown in Table 4, when using autoregressive methods for one-time emotion inference, the 4k window length control group that truncates inputs to accommodate LLM limitations demonstrates weaker emotional understanding capabilities compared to the 32k window length group that can read the entire input at once. This conclusion verifies that complete context information is bene-

Table 4: Performance comparison between models with different context window lengths on the IEMOCAP dataset.

| Model | WA (%) | UA (%) | WF1 (%) |
|---|---|---|---|
| 4k Window | 69.59 | 72.07 | 69.57 |
| 32k Window | 72.85 | 73.34 | 75.58 |

ficial for LLM to understand emotions in long audio, confirming the necessity of DPM inference (understanding complete context information within limited LLM window lengths).

