# OpenReview forum: "Dynamic Parameter Memory: Temporary LoRA-Enhanced LLM for Long-Sequence Emotion Recognition in Conversation"
_ICLR.cc/2026/Conference — ICLR 2026 Conference Withdrawn Submission_

### Official Review · Reviewer_vAs6 · 2025-10-31

**Soundness:** 3
**Presentation:** 3
**Contribution:** 2
**Rating:** 4
**Confidence:** 5

**Summary:**

A speech large language model (SLLM) is proposed for the task of emotion recognition. A DPM mechanism is proposed so that during inference, SLLM can take much longer audio from the conversation for more effective SER.

**Strengths:**

The problem statement and motivation behind the work are well introduced.

The overall presentation of the paper and the visuals are clear.

**Weaknesses:**

For a more comprehensive evaluation, I would suggest more evaluations on more conversational style datasets in addition to IEMOCAP and MELD. This could be something to consider for other language speakers and more varied conversational styles. In addition, a synthetic dataset could also be used to show generalizability.

I would also like to know the model's performance on different discrete emotions, instead of total accuracy or macro F1.

I know the main scope of the work is for ERC, but for SER in general, there are many situations where the datasets are monologues, either natural or scripted. These can have long recordings too. For an ICLR paper, it might not be comprehensive enough to just focus on emotion recognition in conversation.

**Questions:**

None.

---

### Official Review · Reviewer_nJz7 · 2025-10-31

**Soundness:** 1
**Presentation:** 3
**Contribution:** 2
**Rating:** 2
**Confidence:** 4

**Summary:**

The authors in this paper propose a mechanism designated Dynamic Parameter Memory (DPM) to address the critical limitation of a finite context window in Speech Large Language Models (SLLMs) for emotion recognition in long conversations. The core of their approach is to process the audio sequentially, sentence by sentence, and progressively encode the accrued contextual and emotional information into a temporary Low-Rank Adaptation (LoRA) module. This module effectively functions as a dynamic memory that is continuously updated, a design which allows the model to handle audio sequences of theoretically unlimited length with linear computational complexity, as it obviates the need to re-process the entire conversational history. The authors validate their approach on the IEMOCAP dataset, where they report a significant improvement in weighted accuracy and establish a new state-of-the-art performance.

**Strengths:**

1. The primary novelty of this work lies in its innovative application of LoRA not as a static fine-tuning method, but as a dynamic, temporal memory for extending the effective context of an SLLM. Instead of conventional approaches like input compression or sliding windows, which risk losing historical information, the authors propose to progressively encode the evolving conversational context directly into the LoRA parameters during inference. This reconceptualization of LoRA is really interesting for researchers and practitioners in the field that would increasingly try to solve issues with more and more past context. The continuously updated memory buffer is a distinct and compelling approach to circumventing the fixed context window limitations in most of the existing transformer-based models.
2. A significant strength of the proposed Dynamic Parameter Memory (DPM) mechanism is its computational efficiency and inherent scalability for processing long sequences. By adopting a sentence-by-sentence processing scheme that only updates the compact LoRA memory, the method avoids the computationally expensive need to re-process the entire conversational history at each step. This design results in a linear computational cost with respect to the sequence length, making the framework able to get deployed for real-world applications.

**Weaknesses:**

The paper has some weaknesses and I will try to write them down in a somewhat decreasing order of significance that would hopefully help the authors to fix these issues and improve the quality of their paper.

1. A primary concern is the marginal performance improvement when contextualized against the immense increase in model complexity. The reported 10-15% gain in weighted and unweighted accuracy over a four-emotion task on IEMOCAP is unimpressive when compared to results from over seven years ago using simple signal processing features coupled with a two-layer BLSTM architecture proposed more than 7 years ago [A]. It is questionable whether the computational cost is justified (e.g. the 160 million parameters in the LoRA module alone are likely orders of magnitude larger than these older, simpler networks in [A], probably by 2 to 3 orders of magnitude). For the paper's claims to be convincing, the authors must include direct comparisons to these earlier models, or scaled-up versions thereof, to demonstrate that the proposed SLLM-based approach offers a benefit that could not be achieved by more resource-efficient means.
2. Following the previous point, the paper makes claims of efficiency without providing the necessary quantitative evidence. A holistic computational complexity analysis is conspicuously absent. To properly assess whether the performance gains are worth the effort, the authors must explicitly report the key metrics for their model: the end-to-end inference time on their specified CPU hardware, the actual memory footprint required during inference, and the total number of FLOPs. Without this data, it is impossible for the reader to verify the practical viability of the DPM mechanism for on-device or real-time applications and to make a fair comparison of the trade-offs between this complex architecture and simpler, established models.
3. While the proposed method for instilling long-term memory in a LoRA module is intriguing, its application to speech emotion recognition may not be the most compelling use case to demonstrate its capabilities. Given that the performance gains over much simpler recurrent architectures are not dramatic, the problem might not sufficiently require the level of long-context understanding that the DPM is designed to provide. The true potential of this method might be better showcased on tasks where extremely long-term dependencies are unequivocally critical, such as long-form document summarization, narrative video understanding, or context retrieval from extensive texts. In its current form, there is a mismatch between the sophistication of the proposed solution and the demands of the chosen evaluation task.

[A] Tzinis, E., Paraskevopoulos, G., Baziotis, C. and Potamianos, A., 2018. Integrating Recurrence Dynamics for Speech Emotion Recognition. In Proc. Interspeech 2018 (pp. 927-931).

I would gladly increase my score if the authors work properly to address the above issues to a proper degree.

**Questions:**

How can you extend the training for unsupervised learning? Meaning that you can have some random audio recordings that you can only get some initial emotion estimations and you can use those to extend the LoRA adaptation.

---

### Official Review · Reviewer_vBe4 · 2025-11-01

**Soundness:** 3
**Presentation:** 3
**Contribution:** 3
**Rating:** 6
**Confidence:** 3

**Summary:**

Proposes Dynamic Parameter Memory (DPM) — a temporary LoRA-based mechanism for long-sequence emotion recognition in conversations.

Integrates with an emotion-aware Speech Large Language Model (SLLM) to encode sentence-level emotional context.

Enables unlimited-length audio processing under limited context windows with linear complexity.

Achieves state-of-the-art results on IEMOCAP and MELD datasets.

**Strengths:**

1. The manuscript presents a novel inference method (DPM) addressing long-sequence limits in LLMs. This is usually a very complex method that can be very helpful in long sequential emotional recognition conversations.

2. The manuscript also maintains contextual emotion continuity across dialogue turns. This is also very in depth and contextual

3. Demonstrated SOTA performance (e.g., 79.34% WF1). The SOTA performance is a good parameter to consider overall

4. Elegant use of temporary LoRA for dynamic adaptation.

**Weaknesses:**

1. Although the evaluation looks pretty comprehensive but limited to two datasets; lacks real-world or multilingual validation.
2. The metrics are good but there is no explicit latency or computational cost benchmarks.
3. Overall there is a high dependency on sentence segmentation quality.
4. Limited analysis on failure or misclassification cases were also seen overall.

**Questions:**

1. How does DPM handle overlapping speech or noise in real-time audio, this is something that can answered by some metrics ?
2. What is the exact memory footprint and time overhead of per-sentence LoRA updates?
3. Can you test the same framework using a multi agentic MCP approach as well, with applications connected. Idea is to explore real world scenarios?

---

### Note · Authors · 2025-12-09

I have read and agree with the venue's withdrawal policy on behalf of myself and my co-authors.